# The Time-of-Arrival Offset Estimation in Neural Network Atomic Denoising in Wireless Location

**DOI:** 10.3390/s22145364

**Published:** 2022-07-18

**Authors:** Yunbing Hu, Ao Peng, Biyu Tang, Guojian Ou, Xianzhi Lu

**Affiliations:** 1School of Informatics, Xiamen University, Xiamen 361001, China; yunbinghu@stu.xmu.edu.cn (Y.H.); tby@xmu.edu.cn (B.T.); 2Chongqing College of Electronic Engineering, Chongqing 401331, China; ouguojian@cqcet.edu.cn (G.O.); m20092100021@city.edu.mo (X.L.); 3School of Infornation Technology, Xichang University, Sichuan 615000, China

**Keywords:** channel state information, channel estimation, compressive sensing

## Abstract

With the increasing demand for wireless location services, it is of great interest to reduce the deployment cost of positioning systems. For this reason, indoor positioning based on WiFi has attracted great attention. Compared with the received signal strength indicator (RSSI), channel state information (CSI) captures the radio propagation environment more accurately. However, it is necessary to take signal bandwidth, interferences, noises, and other factors into account for accurate CSI-based positioning. In this paper, we propose a novel dictionary filtering method that uses the direct weight determination method of a neural network to denoise the dictionary and uses compressive sensing (CS) to extract the channel impulse response (CIR). A high-precision time-of-arrival (TOA) is then estimated by peak search. A median value filtering algorithm is used to locate target devices based on the time-difference-of-arrival (TDOA) technique. We demonstrate the superior performance of the proposed scheme experimentally, using data collected with a WiFi positioning testbed. Compared with the fingerprint location method, the proposed location method does not require a site survey in advance and therefore enables a fast system deployment.

## 1. Introduction

Due to the emergence of 5G and 6G technology, people have higher and higher requirements for indoor positioning [1,2]. According to Market&Markets research data, the global market size related to indoor location is expected to reach USD 18.74 billion by 2025 [3]. The huge market demand has become a key driver of location-based applications in the future. However, despite the growing demand in a large market, it is not easy to provide a viable indoor positioning solution in many cases. One of the key challenges facing distance-based positioning systems is that the reflection and refraction of wireless signals in indoor environments may lead to a serious decrease in positioning accuracy.

At present, in addition to satellite-based positioning technology [1,2], various positioning technologies are available, such as Bluetooth [4,5], radar [6,7], radio frequency identification [8], ultra-wideband [9], geomagnetic field [10], visible light [11], thermal infrared [12], sound [13], 5G cellular networks [3], and so on. Most of these positioning technologies, however, require additional hardware anchor points and therefore incur additional costs. With the widespread deployment of WiFi systems, it is advantageous to exploit the ubiquitous WiFi [14,15,16] for positioning, which provides broader convenience for the promotion of technology and business cases.

There are currently two CSI-based positioning technologies, which are based on fingerprint and triangulation, respectively. The fingerprint-based localization method needs to collect CSI data at various locations of the deployment site in advance. The FILA approach proposed by Kaishun et al. [17] achieves a 50 percent accuracy at less than 0.6 m in the laboratory. The C-Map proposed by WEN et al. [18] achieves an average positioning error in the comprehensive environment of 0.97 m. CHAUR-HEH et al. [19] adopted a convolutional neural network to localization in a room, which performs the best, resulting in a maximal localization error of 0.92 m with a probability of 99.97%. YUAN et al. In [20], a multi-view discriminant learning approach was developed for indoor localization that exploits both the amplitude and the phase information of CSI to create feature images for each location, and the minimum distance errors for the laboratory and corridor experiments were 0.205 m and 0.109 m, respectively. In [21], the authors proposed to transform the measured data from CSI into images, extract their features, and use deep learning networks to estimate localization, and achieved an accuracy of more than 90% in laboratories. In [22], the authors proposed to transform the measured data of CSI into images and use the classification ability of the convolutional neural network (CNN) for localization, and 70% of the test cases had a localization error under 1.5 m. The fingerprinting-based approach achieves a relatively high positioning accuracy, but requires the deployment site to be surveyed in advance and whenever the environment is changed, which increases the operating costs of commercial applications.

Data fusion-based positioning can achieve a higher positioning accuracy. Zhao et al. [23] proposed a data fusion method of fingerprinting using RSS and CSI data from single access points, which can achieve a positioning accuracy of 1.79 m in a typical laboratory. Li et al. [24] proposed an enhanced particle-filter-based positioning method that combines CSI information and inertial sensor information to achieve an average accuracy of 1.3 m. However, such techniques require more data sources and therefore are not favorable for commercial applications.

Triangulation-based positioning uses angle and/or distance measurements to locate targets. Unlike fingerprinting-based approaches, it does not require data collection in advance, which brings great convenience to commercial deployment. The systems only needs to be updated when the anchor points are moved, which involves surveying the coordinates of anchors only. In [25], Manikanta et al. used multiple measurements of the angle-of-arrival to achieve positioning on standard WiFi equipment, and 60% of the localization error reached approximately one-meter in indoor office deployment. In [14], WiFi devices were located by using the round-trip delay and angle of arrival (AOA) measurements, which require changes to the firmware of target devices. The multiple signal classification (MUSIC) algorithm proposed by Schmidt et al. [26] had a good effect in terms of resolution, estimation accuracy, and stability under the condition of multiple antennas. ArrayTrack [27] proposed by Jie et al. is similar to [26], requiring eight antennas.

This paper proposes a novel algorithm to estimate the TOA from CSI, which enables an accurate localization of standard WiFi devices. The main contributions of this work are as follows:A novel algorithm based on the CS is proposed to analyze the CSI and extract the CIR.A dictionary denoising method based on the direct weight determination of a neural network is derived for denoising CIR signals in compressed sensing. The sparsity of a dictionary is used as the sparse matrix of compressed sensing to estimate the TOA, which improves the accuracy of estimation.The proposed method is validated experimentally and shown to outperform conventional approaches based on MUSIC or estimating signal parameters via rotational invariance techniques (ESPRIT).

The remainder of this paper is organized as follows. The system model, including system architecture and signal structure, is presented in Section 2. The problem-solving process is presented in Section 3. We present the process of solving TOA by the signal reconstruction proposed. Experimental validation is provided in Section 4, followed by concluding remarks in Section 5.

## 2. System Model

In this section, we briefly introduce the localization testbed that we used, and then introduce the experiment data collected with the system.

### 2.1. System Architecture

In this paper, as shown in Figure 1, we used the WiFi-based wireless ad hoc system for positioning (WiFi-WASP) [2,28,29] developed by the Commonwealth Scientific and Industrial Research Organization (CSIRO) of Australia for our experiments. We installed six custom-built WiFi sniffers as anchors for localization in a conference hall and placed an access point (AP) approximately in the middle of the conference hall. The sniffer measures the time stamp of the communication between the sniffers and the AP to estimate the clock offset and offset of the system clock. The sniffers also measure the time stamp of the communication between the target and the AP to locate the target. A computer was placed in the conference hall to collect time stamps collected by sniffers and estimate the location of target devices by using the TDOA. The system is passive and does not interfere with existing standard WiFi systems.

In WiFi-WASP systems, the sniffer clock is post-synchronized by estimating and compensating for clock skew and clock offset to estimate the arrival time of packets based on a common reference clock [30]. This indicates that the locations of the sniffers and the target device is $s_{j}={\left[x_{j},y_{j},z_{j}\right]}^{T}\left(j=1,\ldots ,N\right)$, where $cv={\left[x,y,z\right]}^{T}$ is the coordinate value of the device and *N* is the number of sniffers. Each sniffer in the system has a local clock that is not synchronized with each other or with any external references. Assuming that the target device transmits a packet at $t_{Tx}$, the arrival time of the corresponding radio signal measured by each sniffer is
(1)$$r_{j}=\left(1+\alpha_{j}\right)\left(t_{Tx}+\frac{d_{j}}{c}+\beta_{j}+\Delta t_{j}+m_{j}\right),j=1,\ldots ,J,$$
where $\alpha_{j}$ and $\beta_{j}$ denote the clock skew and clock offset of sniffer *j*, $d_{j}≜\left|s_{j}-cv\right|$ denotes the distance between the target device and sniffer *j*, *c* is the speed of light, $\delta t_{j}$ is the hardware delay (e.g., delay caused by radio frequency (RF) circuitry), and $m_{j}$ is the time measurement error. During system operation, the measured packet arrival time is corrected according to the relevant AGC settings. Since the values of α and Δt are small, in order to directly subtract excessive hardware delay from $r_{j}$, Equation (Equation 1) can be rewritten as
(2)$$r_{j}=\Delta t_{j}+\left(1+\alpha_{j}\right)\left(t_{Tx}+\frac{d_{j}}{c}+\beta_{j}+m_{j}\right),j=1,\ldots ,J,$$

By estimating the clock skew $\alpha_{j}$, clock offset $\beta_{j}$, and the hardware delay $\Delta t_{j}$ of sniffer *j*, as well as the measurement value of $r_{j}$, the measurement value $tr_{j}$ of the synchronous arrival time of sniffer *j* (less than 0.5 s) in a small time can be estimated, which can be expressed as
(3)$$tr_{j}=\frac{r_{j}-\Delta t_{j}}{1+\alpha_{j}}-\beta_{j}-m_{j},$$

Suppose a standard WiFi device transmits two packets in succession at $t_{n}$ and $t_{n+1}$, respectively. $\alpha_{j}$ can be written approximately as
(4)$$\alpha_{j}-\alpha_{k}\approx \frac{r_{j}^{n+1}-r_{j}^{n}}{r_{k}^{n+1}-r_{k}^{n}}-1$$
where *j* and *k* are different sniffers. By assigning one of the sniffers as the reference clock (i.e., $\alpha_{j}$ = 0), the clock skews of other sniffers can be obtained from (Equation 4).

Since the locations of the sniffers are also known, the distance between the transmitter and the sniffer can be readily obtained. Thus, $\beta_{j}$ can be written approximately as
(5)$$\beta_{j}-\beta_{k}\approx \frac{r_{j}}{1+{\hat{\alpha }}_{j}}-\frac{r_{k}}{1+{\hat{\alpha }}_{k}}-\frac{d_{j}}{c}+\frac{d_{k}}{c}$$
where *j* and *k* represent different sniffers and $\hat{\alpha }$ represents the clock skew estimated by (Equation 4).

### 2.2. Signal Structure

The WiFi-WASP system equipment used the orthogonal frequency-division multiplexing (OFDM) modulation scheme. Figure 2 illustrates the system model for obtaining the TOA calculation considered in this paper from OFDM [31]. At the receiving end, the analog data collected by the antenna went through the processes of the analog/digital converter, the serial transmission data were then converted to parallel data, circular prefixes were removed, discrete Fourier transform was conducted [32], and, finally, the frequency domain CSIs were obtained for localization. The CSI in the frequency domain is given by
(6)$$H(f)=\sum_{k=0}^{L-1}\alpha_{k}exp\left(-j2\pi (f_{0}+n△f\right)\tau_{k}),k\in \left[1,L\right],$$
where $f_{0}$ is the central frequency and △f represents the frequency space between adjacent subcarriers. $\alpha_{k}$ and $\tau_{k}$ denote the complex Gaussian channel coefficient and the time delay of the *k*-th signal propagation path, respectively. *L* is the number of multipath components. Since the system adopts an 80 MHz bandwidth, there were 256 subcarriers in total, and the subcarrier space was 80 MHz/256 = 312.5 khz. The number of CSI values in the frequency domain was 245 because 11 subcarriers were unused. *f* denotes the center frequency of each subcarrier. We denote $\tau =\{\tau_{0},\tau_{1},\ldots ,\tau_{L-1}\}$ as the set of true *L* time delays.

Multipath propagation characterized by the channel frequency response (CFR) in the frequency domain signals includes the amplitude–frequency response and phase–frequency response, while, in wireless channels, amplitude attenuation, phase offset, and time delay can be characterized by CIR. In the CS method, assuming that the sparse matrix is an identity matrix and there is only a rotation factor in the sparse matrix, the multiplication of the sparse matrix by the time domain signal is equivalent to performing the Fourier transform. The physical meaning of the sparse matrix is to transform the signal to be sampled into another domain, and, in this domain, the signal is sparse. Thus, we used the CS method to convert the CFR into the CIR in the time domain to estimate the TOA, and we obtained
(7)$$h(\tau )=\sum_{k=0}^{L-1}\alpha_{k}\delta \left(\tau -\tau_{k}\right),k\in \left[1,L\right],$$
where $a_{k}$ and $\tau_{k}$ denote the amplitude of the multipath component and the complex attenuation and propagation delay of the *k*th path, respectively. δ(τ) is the Dirac delta function and *L* is the number of multipath components, while 1≤k≤L are in ascending order. In a line-of-sight (LOS) environment, the first peak of the impulse response is known to be a good choice for the TOA of the direct path.

## 3. Proposed Approach

In this section, the direct weight determination method based on a neural network is introduced to filter dictionary atoms, and the filtering effect is proven by using the quintic polynomial. In order to increase the resolution of TOA, an augmented matrix was added for compressed sensing, and the sparsity and coherence of the proposed method are proven.

Figure 3 illustrates the steps taken to locate a target based on CSI collected by the considered WiFi positioning testbed. A Hamming window was first applied to the CSI to prevent frequency leakage. In order to estimate high-precision TOA, we used neural network weight determination method to filter dictionary atoms. The phase space reconstruction theory was used to reconstruct the input signal to address the problem of insufficient input training signals. The sparsity of dictionary was used as the sparse matrix of compressed sensing, and the resolution of TOA was improved by adding and expanding matrix. However, in a non-LOS or multipath environment, the first arrival path tends to decay significantly and its signal strength may be lower than the reflection path. A peak filtering algorithm was employed to improve the accuracy of estimated TOA. The system clocks among anchors were then synchronized and used to correct the measured TOA values. Finally, the target was located based on the differences between the TOA measured by different anchors.

### 3.1. Frequency Domain Windowing

In the process of digital signal processing, we can only transform the limited length of signal data, so it is necessary to carry out signal truncation. Even if it is a periodic signal, if the truncation time is not an integer multiple of the period, then the intercepted signal will have frequency leakage. To minimize this leakage error, we need to use a weighting function, also known as a window function. There are many kinds of window functions, such as rectangular window, Hann window, and Hamming window. Rectangle window only requires an accurate reading of the main lobe frequency, regardless of amplitude accuracy. Hann window is superior to rectangular window, but the main lobe of Hann window is widened, which is equivalent to the analysis bandwidth broadening, and the frequency resolution decreases. If the test signal has multiple frequency components, the spectrum performance is very complex, and the purpose of the test is more concerned with the frequency points rather than the size of the energy, so Hann window should be selected. Both Hamming window and Hann window are cosine windows, also known as improved ascending cosine windows, but the weighting coefficient is different, which makes the sidelobe smaller, and the sidelobe attenuation speed of the Hamming window is slower than that of Hann window. Due to the fact that the amplitude–frequency characteristic of Hamming window is that the side lobe attenuation is large [33], the peak attenuation of the main lobe and the first side lobe can reach 40 dB. Thus, we have
(8)$$w(l)=\alpha_{0}-\left(1-\alpha_{0}\right)\cdot \cos\left(\frac{2\pi l}{L-1}\right),0\leq l\leq L-1,$$
where $\alpha_{0}$ = 25/46, and this value is intended to generate zero-crossing at the frequency 5π/(L−1). Since the sample point is 256, *L* = 256. Therefore, the CFR data after applying the Hamming window is expressed as
(9)F=H·w,
where $H={\left[h_{1},h_{2},\cdots ,h_{l},\cdots ,h_{L}\right]}^{T}\in C$ is the frequency domain CSI collected by WiFi-WASP system, $w=[w_{0},w_{1},\cdots ,w_{l},\cdots ,w_{L-1}]\in R$ is the Hamming window, and $F={\left[s_{1},s_{2},\cdots ,s_{k},\cdots ,s_{K}\right]}^{T}\in C$.

### 3.2. Dictionary Atomic Filtering

In order to solve the problem of insufficient data of neural network training, the phase space reconstruction theory is proposed to reconstruct signal space. A direct weight determination method based on neural network is proposed to realize dictionary atomic filtering, and its effectiveness is verified.

#### 3.2.1. The Phase Space Reconstructs the Input Signal

The theory of phase space reconstruction holds that the prime mover system model can be reconstructed from an observation of the system. The evolution of any component in the system is determined by the other components interacting with it, so the information of these related components implies that, in the development process of any component, a spatial system model of the original signal can be reconstructed with an observation of the system.

Due to the fact that each set of signals sampled is different, there are too little data to train the dictionary. According to the phase space reconstruction theory [34,35], for the noise signal $F\in C^{K\times 1}$ sampled from WiFi-WASP system and through the Hamming window, it can be embedded into the attritor orbit matrix F in dimension *K* so as to solve the problem of insufficient training data F.
(10)$$F=\left[\begin{array}{cccc} s_{1}, & s_{2},\; & \cdots , & s_{K-N+1}\\ s_{2}, & s_{3},\; & \cdots , & s_{K-N+2}\\ \vdots & \vdots & \ddots & \vdots \\ s_{N}, & s_{N+1},\; & \cdots , & s_{K} \end{array}\right],$$
where $F\in C^{N\times K}$.

#### 3.2.2. Dictionary Learning Algorithms

The dictionary learning model is widely concerned and has been successfully applied in various fields of signal processing, such as signal processing, image processing, image fusion, video key frame extraction, and so on. In the field of signal processing, sparse representation of signals is a simple and effective signal coding method, which describes signals by using linear combinations of as few basis vectors as possible in the basis space. The span of basis vectors is also called a dictionary, and the basis vectors in the corresponding space are atoms.

The dictionary learning algorithm is also very suitable for signal recovery and reconstruction and image noise and blur removal applications. For the dictionary learning algorithm of signal denoising, the trained data can be divided into two categories [36]. First, the method that uses data without noise for dictionary learning is called the external prior dictionary learning algorithm. Second, the method of using noisy data for dictionary learning is called the internal prior dictionary learning algorithm. The external prior dictionary learning algorithm is not adaptive to signal denoising, and it may not be able to reconstruct some fine-scale noise signals well. Compared with the external prior dictionary learning algorithm, the internal prior dictionary learning algorithm adopts the signal data containing noise as the training data; although it has good adaptability, the prior learning is greatly affected by noise. The denoising effect of using such a dictionary to denoise signals is inevitably lower than that of using a fixed atom dictionary. Therefore, reducing the atomic noise of a learning dictionary will definitely improve the denoising effect of signals. The main idea of dictionary learning is to use the dictionary matrix $\Psi \in R^{N*K}$, which contains *K* atoms $\psi_{k}$ and a sparse linear representation of the original sample $Y\in R^{N*L}$, where m represents the number of samples, n represents the attributes of samples, and, in the ideal case, Y=ΨX. This can be transformed into an optimization problem.
(11)$$\begin{array}{cc} \; & \min_{\Psi ,X}{∥Y-\Psi X∥}_{F}^{2},\\ \; & s.t.\forall i,∥X_{i}{∥}_{0}\leq T_{0}, \end{array}$$
where $X\in R^{K\times L}$ is the sparse decomposition coefficient of the signal, and $X_{i}$ is the row vector in the matrix. $∥X_{i}{∥}_{0}$ represents the zero-order norm, which represents the number of non-zero numbers in a vector. $T_{0}$ represents the maximum sparsity of $∥X_{i}{∥}_{0}$. There are two optimization variables Ψ and X to solve this optimization problem. It is common to fix one optimization variable, optimize the other variable, and so on. The sparse matrix X in Formula (Equation 11) is solved by the least absolute shrinkage and selection operator classical algorithm.

#### 3.2.3. Direct Weight Determination of Neural Networks

Neural networks with their massively parallel processing, distributed storage, and elastic topology, and other significant characteristics of highly redundant and nonlinear arithmetic, such as the control and nonlinear signal processing, pattern recognition, and robots, have been widely used in such fields as [37]. According to the model of the artificial neural network, the structure can be divided into a feedforward network (also known as a multilayer perceptron network) and the feedback network (also known as the Hopfield net). Among neural network models, the back propagation (BP) neural network model is one of the most widely used neural network models at present. Its core is the error back propagation algorithm, and its model is shown in Figure 4.

The model shows a three-layer BP neural network model that includes an input layer, hidden layer, and output layer. The input layer contains *N* neurons, so the corresponding input $I_{i}=\left[x_{1},x_{2},\cdots ,x_{n},\cdots ,x_{N}\right]$ is an *N* dimension vector. There are *L* neurons in the hidden layer and the excitation function is denoted as $f_{1}$. In addition, the expected output $O_{i}$ is an I dimensional vector and $\omega_{nl}$ is the connection weight between the *n* neuron of the input layer and the *L* neuron of the hidden layer, whereas $\varpi_{lj}$ is used to represent the connection weight between the *l* neuron of the hidden layer and the *j* neuron of the output layer. $\vartheta_{l}$ and $\theta_{j}$ are used to represent the threshold value of the neuron at the hidden layer and the neuron at the output layer, respectively.

$\nu_{1}$ denotes the first *l* hidden layer neurons of the output, and $y_{j}$ represents the actual output of the *j* neuron in the output layer, $O_{i}=\left[O_{i1},O_{i2},\cdots ,O_{ij},\cdots ,O_{IJ}\right]$ for the sample output. According to the direction of negative gradient descent, the BP algorithm iteratively adjusts the weights and thresholds of the network to achieve the reduction in the error function value, and its weight $\omega_{nl}$ and $\varpi_{lj}$ correction iterative formula can be described as
(12)$$\left\{\begin{array}{cc} \; & \omega_{nl}⟵\omega_{nl}+\Delta \omega_{nl}\\ \; & \varpi_{lj}⟵\varpi_{lj}+\Delta \varpi_{lj}, \end{array}\right.$$
where
(13)$$\left\{\begin{array}{cc} \Delta \varpi_{lj} & =-\eta \frac{\partial E}{\partial \varpi_{lj}}=-\eta \frac{\partial E}{\partial y_{j}}\cdot \frac{\partial y_{j}}{\varpi_{lj}}=\eta \left(O_{ij}-y_{j}\right)\cdot f_{2}^{\prime }\cdot v_{l}\\ \Delta \omega_{nl} & =-\eta \frac{\partial E}{\partial \omega_{lj}}=-\eta \frac{\partial E}{\partial y_{j}}\cdot \frac{\partial y_{j}}{\partial v_{l}}\cdot \frac{\partial v_{l}}{\partial \omega_{l}}\\ \; & =\eta \sum_{j=1}^{J}\left(O_{ij}-y_{j}\right)\cdot f_{2}^{\prime }\cdot \varpi_{lj}\cdot f_{1}^{\prime }\cdot x_{n}. \end{array}\right.$$

In the above formula, $v_{l}$ represents the output of the *l*th neuron, $y_{j}$ represents the output of the *j*th neuron, and *E* represents the neural network learning error function corresponding to the *j*th sample pair under the incremental processing mode. Their derivation formulas are as follows:(14)$$\left\{\begin{array}{cc} \; & v_{l}=f_{1}\left(\sum_{n=1}^{N}\omega_{nl}x_{n}-\vartheta_{l}\right),l=1,2,\cdots ,L\\ \; & y_{j}=f_{2}\left(\sum_{l=1}^{L}\varpi_{lj}v_{l}-\theta_{j}\right),j=1,2,\cdots ,J\\ \; & E=\frac{1}{2}\sum_{j=1}^{J}{\left(y_{j}-O_{ij}\right)}^{2}, \end{array}\right.$$

The BP network has some problems, such as a slow convergence, local minimization, and fixed learning rate. To solve these problems, Zhang yu-nong et al. proposed an improved power-excited forward neural network model by using polynomial interpolation and approximation theory [38]. Different from the traditional idea of obtaining neural network weights through lengthy BP iterative training, the direct weight determination method can directly obtain the weights, greatly shortening the determination time of network weights. It overcomes the inherent defect that the traditional iterative training algorithm is susceptible to a dynamic learning process and that it is difficult to determine the optimal topology of the neural network. This model can still be regarded as a BP neural network model, as shown in Figure 5. In this figure, the number of hidden layer neurons is *n*, and the connection weight between hidden layer neurons and output layer neurons is denoted as $\varpi_{j},j=0,1\cdots ,n-1$, but the relation between the input *x*and the output *y* of the neural network can be obtained as $y=\varpi_{0}x^{0}+\varpi_{1}x^{1}+\varpi_{2}x^{2}+\cdots +\varpi_{n-1}x^{n-1}$.

In the figure, the sample set $(x_{i},y_{i}),i=1,2,\cdots ,n$ is used to train the power-excited forward neural network, and the learning error function *E* is defined as $E=\frac{1}{2}\sum_{i=1}^{n}\left(y_{i}-\sum_{p=0}^{n-1}\varpi_{p}x_{i}^{p}\right)$. Therefore, the weight iterative formula can be expressed as:(15)$$\begin{array}{cc} \varpi_{j}\left(k+1\right) & =\varpi_{j}\left(k\right)-\eta \frac{\partial E}{\partial \varpi_{j}}\\ \; & =\varpi_{j}\left(k\right)-\eta \sum_{i=1}^{m}x_{i}^{j}\left[\left(\sum_{p=0}^{n-1}\varpi_{p}x_{i}^{p}-y_{i}\right)\right], \end{array}$$
where j=[1,2,⋯,n−1]. Let $\varpi ={\left[\varpi_{0},\varpi_{1},\cdots ,\varpi_{n-1}\right]}^{T}\in R^{n}$ and $y={\left[y_{0},y_{1},\cdots ,y_{n-1}\right]}^{T}\in R^{m}$, respectively, where the superscript *T* expresses the matrix transpose operation. Let X for
(16)$$X=\left(\begin{array}{cc} x_{1}^{0}\; & x_{1}^{1}\;x_{1}^{2}\;\cdots \;x_{1}^{n-1}\\ x_{2}^{0}\; & x_{2}^{1}\;x_{2}^{2}\;\cdots \;x_{2}^{n-1}\\ \; & \ddots \;\;\;\;\;\;\vdots \\ x_{m}^{0}\; & x_{m}^{1}\;x_{m}^{2}\;\cdots \;x_{m}^{n-1} \end{array}\right)\in R^{m\times n}.$$

Thus, Formula (Equation 15) can be expressed in matrix vector form as follows:(17)$$\begin{array}{c} \varpi \left(k+1\right)=\varpi \left(k\right)-\eta X^{T}\left[X\varpi \left(k\right)-y\right]. \end{array}$$

For Formula (Equation 15), when the network training reaches the steady state, we have
(18)$$\begin{array}{c} \lim_{k\to +\infty }\varpi \left(k+1\right)=\lim_{k\to +\infty }\varpi \left(k\right)=\varpi , \end{array}$$

Then, there is $-\eta X^{T}\left[Xw-y\right]=0$. The optimal weight of the power-excited neural network can be directly determined as ϖ by using the matrix pseudo-inverse idea as follows:(19)$$\begin{array}{c} \varpi \left(k\right)={\left(X^{T}X\right)}^{-1}X^{T}y. \end{array}$$

It can be seen from the above that the direct weight determination method can automatically, quickly, effectively, and accurately determine the optimal weight, so as to achieve network performance optimization.

#### 3.2.4. Dictionary Atomic Denoising Verification

Regardless of using the recursive least squares dictionary learning algorithm (RLS-DLA), K-singular value decomposition (K-SVD)K-SVD, or other dictionary learning algorithms, as long as the training data contain noise, the resulting dictionary is bound to contain noise. In order to prove that the dictionary atom trained by RLS-DLA contains noise and verify the effect of dictionary learning by the direct weight determination method, we set the real quintic polynomial phase signal as the input signal, which can be expressed as
(20)$$\begin{array}{c} y\left(n\right)=A\left(n\right)sin(\sum_{i=0}^{5}a_{i}{\left(\Delta n\right)}^{i})+v\left(n\right),n\in \left[0,N\right], \end{array}$$
where a(n)=1. v(n) is Gaussian white noise, and the signal-to-noise ratio was set to 10 dB. In order to avoid the ambiguity of $a_{i}$, it follows $|a_{i}|\leq \frac{\pi }{m!n^{m-1}\Delta^{m}}$, so we have $a=\left(a_{5},a_{4},a_{3},a_{2},a_{1},a_{0}\right)=\left(4e-10,4e-9,2.75e-6,1.5e-3,\pi /8,\pi /3\right)$.

According to the algorithm idea of a neural network, a dictionary learning algorithm for denoising polynomial phase signals based on the direct weight determination method is proposed. $D\in R^{N\times L}$ initializes the dictionary. Each column in the dictionary represents an atom, and *L* represents the number of atoms in the dictionary. Since a redundant dictionary is used, L≫N. $Y\in R^{N\times K}$ represents the training set, where K≫L; thus, the coefficient matrix $W\in R^{L\times K}$. Since the denoising signal itself is used as training data, the observation length of the signal needs to be considered. In the processing of polynomial phase signals, the signal length is not very long, and the number of trained signals *K* in the training signal set Y is much larger than the number of atoms in the redundant dictionary. Therefore, if the length of the object signal y(i)(i=1,2,⋯,M) is not long enough, in order to satisfy K≫L, the training signal set Y is reconstructed by referring to the phase space reconstruction theory, which can be expressed as
(21)$$Y=\left[\begin{array}{cccc} y_{1}, & y_{2},\; & \cdots , & y_{N-K+1}\\ y_{2}, & y_{3},\; & \cdots , & y_{N-K+2}\\ \vdots & \vdots & \ddots & \vdots \\ y_{K}, & y_{K+1},\; & \cdots , & y_{N} \end{array}\right].$$

At the same time, any observed signal in segment *K* is taken as the initial dictionary D, and the *i*th atom $d_{i}={\left[d_{i1},d_{i2},\cdots ,d_{iN}\right]}^{T}$ of dictionary D is obtained. The input excitation matrix X can be expressed as
(22)$$X=\left[\begin{array}{cccc} x_{1}^{0}, & x_{1}^{1},\; & \cdots , & x_{1}^{n-1}\\ x_{2}^{0}, & x_{2}^{1},\; & \cdots , & x_{2}^{n-1}\\ \vdots & \vdots & \ddots & \vdots \\ x_{N}^{0}, & x_{N}^{1},\; & \cdots , & x_{N}^{n-1} \end{array}\right].$$According to the neural network weight determination method, $w={\left(X^{T}X\right)}^{-1}X^{T}d_{i}$. Therefore, the predicted atomic value after filtering can be obtained as $d_{i}^{*}=Xw$, and the filtered dictionary can be expressed as $D^{*}=\left[d_{1}^{*},d_{2}^{*},\cdots ,d_{i}^{*}\cdots ,d_{L}^{*}\right]$.

As shown in Figure 6, it can be seen from the figure that the real quintic polynomial phase signal with additive Gaussian white noise is a smooth curve after learning by the direct weight determination method. The dictionary atom after RLS-DLA learning obviously contains additive white Gaussian noise. Therefore, the dictionary composed of atoms learned by the direct weight determination method is denoised by sparse representation, and the denoising effect is very obvious.

### 3.3. Sparse Blind the TOA Offset Estimation

In this section, sparse dictionary learning is used to represent signals sparsely. Meanwhile, in order to improve the resolution of TOA, the signal aug-expanding matrix is added and its sparsity and coherence are proven.

#### 3.3.1. Construction of Sparse Basis Matrix

Sparsity is the premise of the compressed sensing algorithm and the necessary condition for signals. However, most signals do not meet the conditions of sparsity or compressibility. In this case, signals need to be transformed into sparse signals in a certain transform domain, and the transformed signals become compressible signals. The signal $ℜ(F)={\left[\varsigma_{1},\varsigma_{2},\cdots ,\varsigma_{l},\cdots ,\varsigma_{L}\right]}^{T}\in R$ sampled by us was a discrete signal in the frequency domain with dimension *L*. The dictionary Ψ estimated by the direct weight determination method was used to estimate the sparse representation coefficient α as the sparse matrix. The signal ℜ(F) can be expressed as
(23)ℜ(F)=Ψα,
where $\Psi \in R^{L\times L}$, and L=256 is the number of dimensions of F of the sampled signal. $\alpha ={\left[\alpha_{1},\alpha_{1},\cdots ,\alpha_{l},\cdots ,\alpha_{L}\right]}^{T}$ is the coefficient vector, which is another representation of F and is sparse on $\Psi^{L\times L}$.

#### 3.3.2. Construction of Measurement Matrix

In the whole process of compressed sensing, the design of the measurement matrix is a key step. The properties of the measurement matrix are related to whether the compression can be achieved and whether the signal can be reconstructed accurately. How can an appropriate observation matrix be designed, which can not only achieve the purpose of compressed sampling but also ensure that the signal can be reconstructed without distortion? The restricted isometry property (RIP) can solve this problem in theory [39]. The random Gaussian measurement matrix is the most widely used in compressed sensing. We constructed a matrix Φ with the size of K*L, and each element in the matrix independently followed a Gaussian distribution with mean value υ=0 and variance of $\sigma =\sqrt{1/K}$, which can be expressed as follows
(24)$$\Phi_{k,l}∼N\left(\upsilon ,\sigma \right),l\in \left[1,L\right],k\in \left[1,K\right],$$
where *L* is the dimensions of the sampled data, and *N* is the normal distribution function. *K* is the number of CIR dimensions to be restored. The random Gaussian measurement matrix has a strong randomness. It can be proven that, when the measurement number K≥cMlog(L/M) of the random Gaussian measurement matrix, RIP conditions will be met with a great probability. In general, there is M<<K≤L, and *c* is a tiny constant. In the compressed sensing process, a random Gaussian measurement matrix is widely used mainly because it is not related to most orthogonal bases or orthogonal dictionaries, and the number of measurements required for accurate reconstruction is relatively small.

#### 3.3.3. CIR Resolution Augmentation Matrix

In order to improve the time domain resolution and recover CIR to estimate a more accurate TOA, we constructed a matrix to extend the arrival delay τ [40], which can be expressed as
(25)$$S_{nk}=sinc\left(\pi \left(A_{nk}\right)/T\right),n\in \left[1,N\right],k\in \left[1,K\right],$$
where sinc is a filter constructed by the sinc function, often used in anti-aliasing techniques. *T* is the sampling period. *N* is the resolution of CIR of data to be restored, and *N* is the control parameter of the resolution of data to be restored, which can be used to improve the resolution of data to be restored, and N>K. $A^{N\times K}\in R$ can be represented as
(26)$$\begin{array}{c} A=\left[\begin{array}{ccccc} \; & 0, & \nu_{1,1}, & \;\cdots , & \nu_{1,k-1}\\ \; & -\nu_{2,2}, & -\nu_{2,1}, & \;\cdots , & \nu_{2,k-3}\\ \; & -\nu_{3,4}, & -\nu_{3,3}, & \;\cdots , & \nu_{3,k-5}\\ \; & \ddots & \vdots \\ \; & -\nu_{n,2*(n-1)}, & \nu_{n,2*(n-1)-1}, & \;\cdots , & \nu_{n,k-2*(n-1)-1} \end{array}\right], \end{array}$$
where ν is the sampling interval and $S^{N\times K}\in R$.

#### 3.3.4. Sparse Recovery

The K*L dimension matrix formed by the product of measurement matrix $\Phi^{K\times L}$ and transformation basis $\Psi^{L\times L}$ is called the perception matrix, and observes the signal and obtains the observation vector $y={\left[y_{1},y_{2},\cdots ,y_{n},\cdots ,y_{N}\right]}^{T}\in R$. The observed signal can be expressed as
(27)y=Φℜ(F)=ΦΨα.

Meanwhile, in order to improve the resolution, we constructed a matrix $S^{N\times K}$ to extend the arrival delay, so (Equation 27) can be expressed as
(28)y=SΦℜ(F)=SΦΨα.

In the context of CS, the CIR reconstruction problem is now formulated as
(29)$$\begin{array}{cc} \; & \min_{\alpha }{∥\alpha ∥}_{0},\\ \; & {s.t.∥y-S\Phi \Psi \alpha ∥}_{2}^{2}\leq \varepsilon , \end{array}$$
where, for some small ε, ${∥\cdot ∥}_{0}$ denotes $l_{0}$-norm, which counts the number of non-zero elements, and ${∥\cdot ∥}_{2}$ stands for $l_{2}$-norm. Note that ε approximates $\sigma^{2}$ if we have enough data. However, $l_{0}$-norm minimization is a complex combinatorial problem [41]. In order to improve the computational efficiency, it is usually extended to $l_{1}$-norm minimization, because it can be solved by convex optimization [42,43]. Compared with the inverse Fourier transform, this CS-based approach enhances sparsity and effectively minimizes the pseudo-sidelobe.

#### 3.3.5. CS with Complex-Valued Targets

The CS estimates a sparse signal vector in a probabilistic manner. It maximizes the posterior probability of the sparse signal vector at a given observed value. In the posterior probability maximization, the CS theory is applicable to the processing of real data, while CSI data are complex. In order to apply this theory to TOA estimation, observation data should be realizable in order to build an optimization model of a real number field [44]. For the application of CS in complex valued CIR estimation, Equation (Equation 28) is first extended to the equivalent real-valued form by
(30)$$\begin{array}{cc} \tilde{H}\; & =\left[\begin{array}{c} ℜ(\Phi \Psi S)\;-ℑ(\Phi \Psi S)\\ ℑ(\Phi \Psi S)\;\;ℜ(\Phi \Psi S) \end{array}\right]*\left[\begin{array}{c} ℜ(\alpha )\\ ℑ(\alpha ) \end{array}\right]\\ \; & =\tilde{S}\tilde{\Phi }\tilde{\Psi }\tilde{\alpha }, \end{array}$$
where $\tilde{H}\in R^{2N\times 1}$, $\tilde{\Phi }\in R^{2K\times 2L}$, $\tilde{\Psi }\in R^{2L\times 2L}$, $\tilde{S}\in R^{2N\times 2K}$, and $\tilde{\alpha }\in R^{2L\times 1}$. This expansion increases the dimension of the observation matrix and separates the reconstruction of real and imaginary values of a channel magnitude in each delay tap.

#### 3.3.6. Sparse and Coherence Proof

In the whole process of compressed sensing, the design of a measurement matrix is a key step. The properties of the measurement matrix are related to whether the compression can be achieved and whether the signal can be reconstructed accurately. For a given measured value *y*, the solution of Formula (Equation 28) is an underdetermined problem. Candes et al. proposed a solution: as long as the measurement matrix conforms to the RIP, a definite solution can be obtained; that is, the existence, accuracy, and uniqueness of the original signal reconstruction can be guaranteed. RIP is defined as:(31)$$\begin{array}{c} {\left(1-\delta \right)∥x∥}_{2}^{2}\leq {∥x∥}_{2}^{2}\leq \left(1+\delta \right){∥x∥}_{2}^{2}, \end{array}$$
where δ is a constant and δ∈(0,1). ${∥\cdot ∥}_{2}$ stands for $\ell_{2}$-norm. In this paper, the measurement matrix is a random Gaussian measurement matrix, and there is a high probability that RIP conditions are met.

According to Baraniuk, the equivalent condition of RIP is that the observation matrix Φ and the sparse representation basis Ψ are irrelevant. However, since we added the S matrix to improve the CIR resolution, the equivalent condition of RIP is therefore irrelevant between Φ, Ψ and S. In practical application, we can use an equivalent condition of criterion, namely incoherence, to guide the design of the measurement matrix. Incoherency means that the row vector $\Phi_{i}$ of the matrix observation system Φ cannot be sparsely represented by the column vector $\Psi_{i}$ in the basis Ψ, nor can it be represented by the column vectors of the $S_{i}$ matrix. It can be measured by coherence and is expressed as
(32)$$\begin{array}{c} \mu \left(\Phi ,\Psi ,S\right)=\max_{1\leq k\leq K,1\leq l\leq L,1\leq n\leq N}|\langle \Phi_{k},\Psi_{l},S_{n}\rangle |. \end{array}$$

In this paper, $\Phi^{K\times L}$ is a sensing matrix of random normal distribution with mean υ=0 and variance $\sigma =\sqrt{1/K}$. According to the normal distribution function, the ratio within $3\sqrt{1/K}$ is approximately 99.73%, so
(33)$$0\leq |\Phi_{kl}^{K\times L}{|}_{1}≲3\sqrt{1/K},k\in \left[1,K\right],l\in \left[1,L\right].$$

The sparse matrix Ψ estimated by dictionary learning is derived from collected CSI data. The indoor location signal model of the wiFI-WASP system used to collect CSI data is suitable for the Saleh–Valenzuela propagation model. The Saleh–Valenzuela propagation model is an indoor radio channel statistical multipath model that fits well with our measurements, through which, the received signal rays arrive in clusters. These rays have an independent uniform phase, as well as independent Rayleigh amplitudes whose variances decay exponentially with the cluster and ray delay. Light in clusters and within clusters forms Poisson arrival processes with different but fixed rates. According to the model, the received signal amplitude follows a normal distribution, which can be expressed as
(34)$$\;a_{k}=N\left(0,\sigma_{k}^{2}\right)+jN\left(0,\sigma_{k}^{2}\right),$$
where *k* indicates the number of paths and $N(0,\sigma_{k}^{2})$ shows a value that is in accordance with the normal distribution. Equation (Equation 34) creates the Rayleigh fading. The variance $\sigma_{k}^{2}$ is the average power of the k path, so the strength of the paths within the clusters is given as [45,46]:(35)$$\sigma^{2}\left(k\right)=e^{-T_{l}/\Gamma }e^{{-\tau }_{il}/\gamma },$$
where $T_{l}$ is the arrival time of the *l*th cluster and $\tau_{il}$ is the arrival time of the *i*th path in the *l*th cluster. We set the time constants $T_{l}$ and $\tau_{il}$ or the inter and intra cluster as 300 ns and 5 ns, respectively, as Poisson distributions. Γ is a constant of cluster arrival decay time, γ indicates a constant of ray arrival decay time, Γ = 60 ns, and γ = 20 ns. According to the normal distribution function, the ratio within $3\sqrt{1/\sigma^{2}\left(k\right)}$ is approximately 99.73%. Thus, it can be seen that the range of $l_{1}$ norm of any element in Ψ can be expressed as
(36)$$0\leq |\Psi_{uv}^{L\times L}{|}_{1}≲3\sigma .$$

The sinc function of CIR resolution enhancement matrix $S^{N\times K}$ can be expressed as
(37)$$S_{nk}=sinc\frac{\pi A_{nk}}{T}=\frac{sin(\left(\pi A_{nk}\right)/T)}{\pi A_{nk}/T},n\in \left[1,N\right],k\in \left[1,K\right].$$

Obviously, the value range of any element in matrix S is
(38)$$0\leq |S_{nk}{|}_{1}\leq 1.$$

Therefore, the coherence of Φ, Ψ, and S when multiplied by any rows and columns is in the range of
(39)$$\begin{array}{c} 0<\mu \left(\Phi ,\Psi ,S\right)≲\frac{9L\sigma }{\sqrt{K}}, \end{array}$$
where the coherence of μ is closely related to the number of rows and columns of Φ and Ψ, but not to the number of rows and columns of S and σ≪1. Since the elements in the S matrix are generally less than 1, it is still helpful for incoherence μ.

## 4. Experimental Results

The performance of the proposed algorithm was evaluated under outdoor line-of-sight (LOS) conditions and indoor conditions, respectively. In each experiment, an 802.11ac wireless LAN was built, and six sniffers were deployed, running in 149 channels with a bandwidth of 80 MHz. One target device was used for location verification and one PC was used to collect data from the sniffer and target device online to estimate the location of the target device.

In practical applications with LOS propagation, usually the first peak of the impulse response is the TOA of the direct path. However, in a non-LOS or multipath environment, the first-arrival path usually attenuates significantly and may even have a lower signal intensity than the reflection path. In addition, multiple reflected signals that overlap the direct path signal distort the first peak of the combined signal [10]. Existing ToA estimation algorithms are based on thresholding the first peak, using a prior model [3], or the MUSIC/ESPRIT algorithm. The performance of the proposed approach was compared against MUSIC and ESPRIT.

### 4.1. Outdoor Test

The system was tested under outdoor LOS conditions to evaluate its performance. We deployed six sniffers around a WiFi network for the outdoor LOS tests with one AP and one WiFi dongle. The topology of the system is shown in Figure 7. The size of the experimental area was 900 m${}^{2}$. The target moved across the test area. Figure 5 shows the positioning results for the outdoor test. It can be seen that the positioning results are consistent with the actual locations.

Figure 8 shows the positioning accuracy of our system when the TOA is estimated using the CS and MUSIC and ESPRIT algorithm. It can be seen that the 72-percentile positioning error is 0.5 m, and 1 m in 90% with the CS. Compared with MUSIC and ESPRIT’s TOA estimation methods, the accuracy of CS’s TOA estimation method is much higher.

### 4.2. Indoor Test

Localization accuracy depends on the multipath environment, the materials used in the walls, the presence of metal objects, the density of sniffer deployment, and many other factors, so indoor localization testing is an important step in validating wireless positioning methods. The assembly hall covers an area of around 70 m${}^{2}$. We deployed six sniffers and a standard Wi-Fi device in the conference hall and placed them at known locations to collect data from 17 locations of target devices inside the conference hall. Figure 9 shows the real and estimated location of the target. It can be seen that, except for the poor reflection effect of the seat in the Y-axis direction, the estimated location is consistent with the actual location, which can meet the needs of many indoor positions.

Figure 10 shows the positioning effect when using CS, MUSIC, and ESPRIT algorithms to estimate the TOA. We observed that CS achieves a localization error of approximately the 48th percentile at 0.5 m and 1 m, respectively. Compared with MUSIC and ESPRIT’s TOA estimation method, CS’s TOA estimation method has a higher accuracy. The SpotFi positioning system is an indoor localization method proposed by m. Kotaru et al. The SpotFi localization system has three antennas and is deployed in the indoor office with 5–6 AP. The localization error of approximately 18% is 0.5 m and approximately 60% is 1 m [25]. The ArrayTrack localization system is an indoor localization method proposed by Jie Xiong et al. In the same experimental environment as SpotFi, the localization error of approximately 30 % is 1 m.

## 5. Conclusions

With the increasing demand for positioning, it is difficult for existing positioning methods to meet the convenience of deployment equipment without satellite positioning. The proposed TOA estimation method is expected to better adapt to the business requirements of equipment deployment. In this paper, a method based on a neural network to determine the direct weight of dictionary filtering was designed, and the filtering effect is proven by quintic polynomial. The learned dictionary was used as the sparse matrix of compressed sensing, the augmentation and expansion matrix was added to increase the CIR resolution after sparse recovery, and the sparsity of the dictionary, measurement matrix, augmentation, and expansion matrix was proven. The proposed algorithm was validated by experiments on the indoor and outdoor positioning system. The results show that the proposed algorithm has a high positioning accuracy. Compared with the fingerprint positioning method, the positioning method adopted in this paper has the advantage of not needing to sample the data in the positioning environment in advance, so it has a broad application prospect.

## Figures and Tables

**Figure 1 sensors-22-05364-f001:**
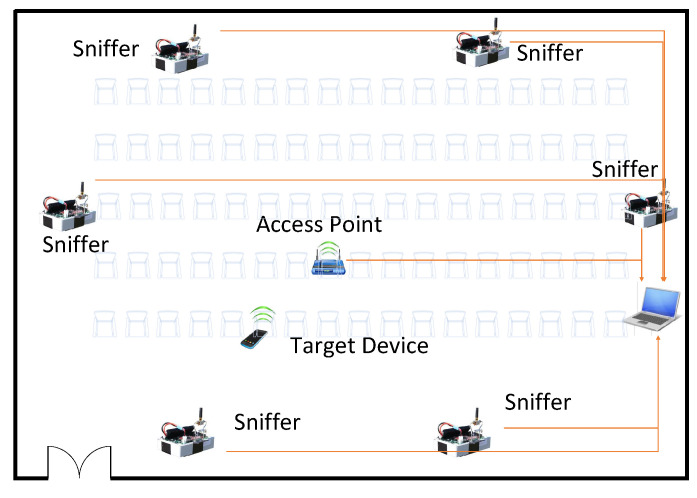
Structure of the TDOA-based passive WiFi localization system.

**Figure 2 sensors-22-05364-f002:**
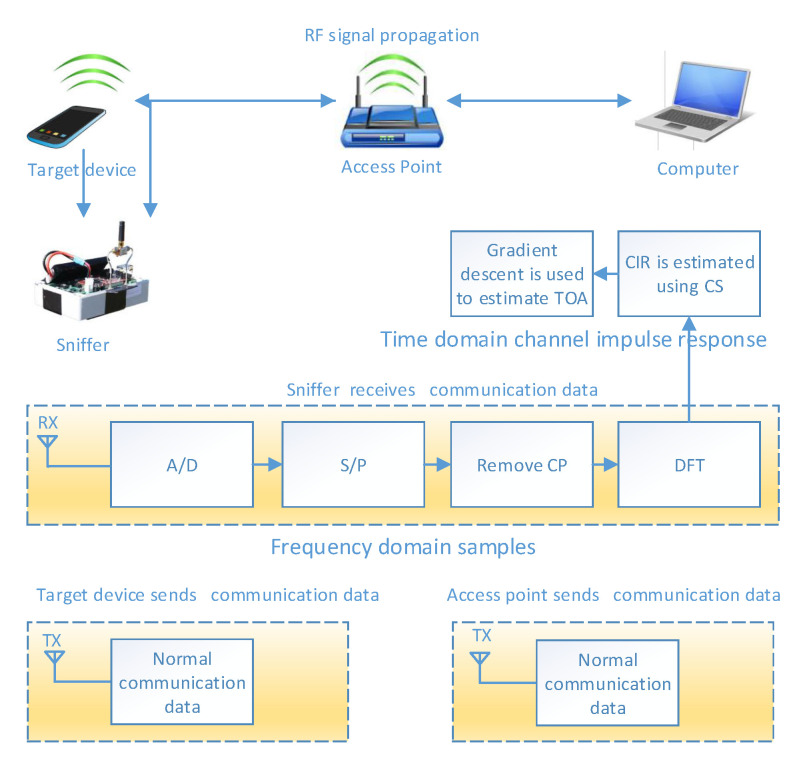
System model for the TOA computation in WiFi-WASP localization using OFDM.

**Figure 3 sensors-22-05364-f003:**
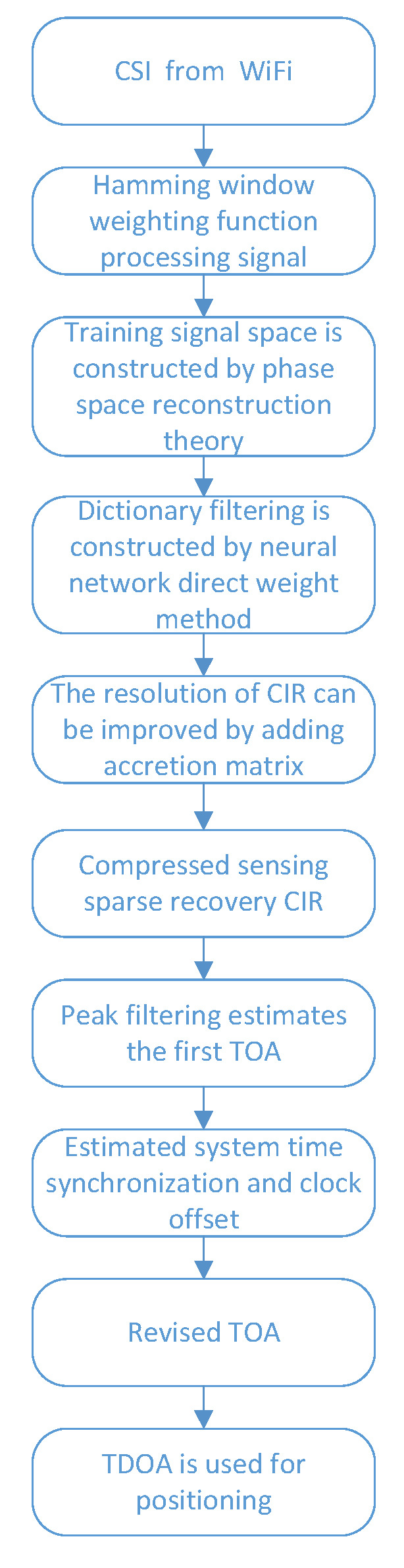
Algorithm framework is based on CSI localization.

**Figure 4 sensors-22-05364-f004:**
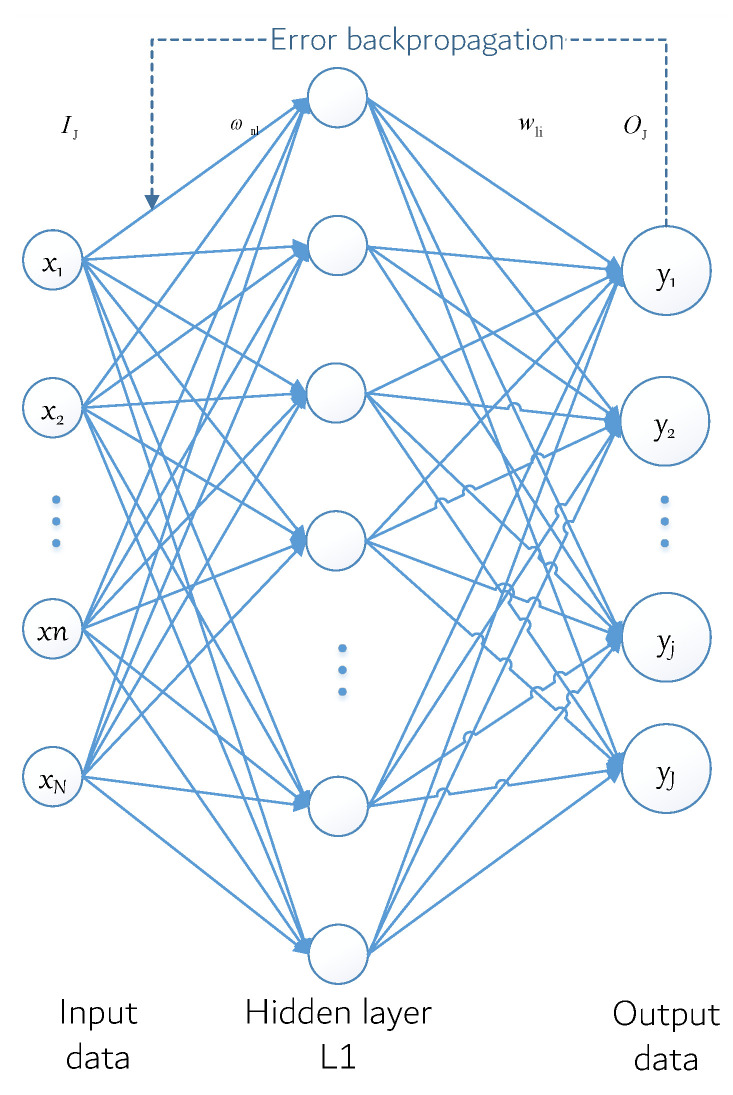
BP neural network model.

**Figure 5 sensors-22-05364-f005:**
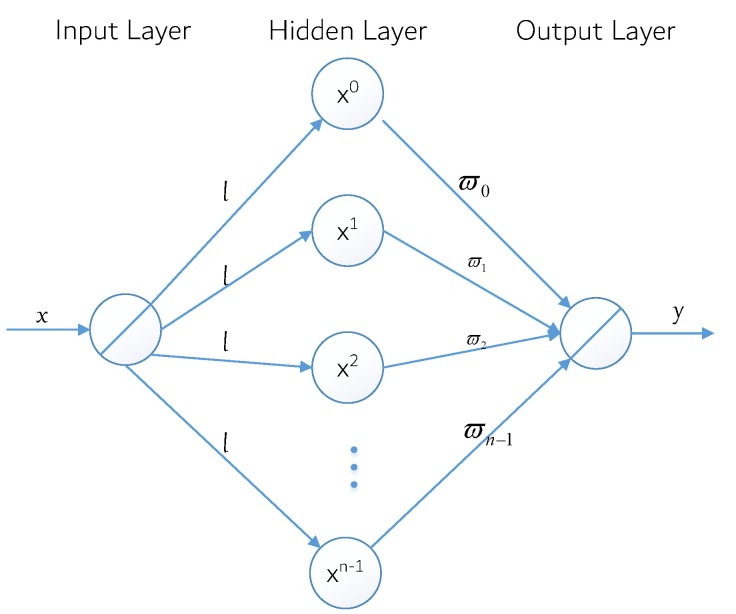
An improved power-excited forward neural network model.

**Figure 6 sensors-22-05364-f006:**
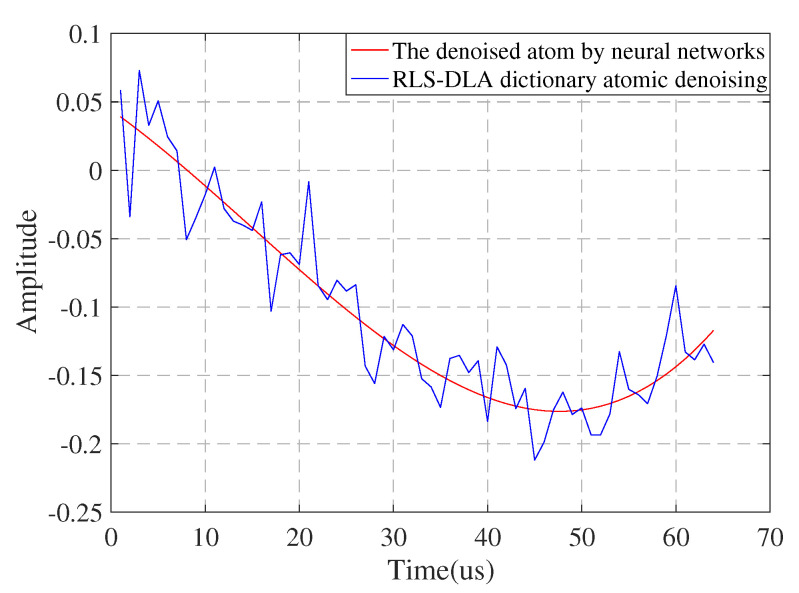
Comparison of neural network and RLS-DLA denoising.

**Figure 7 sensors-22-05364-f007:**
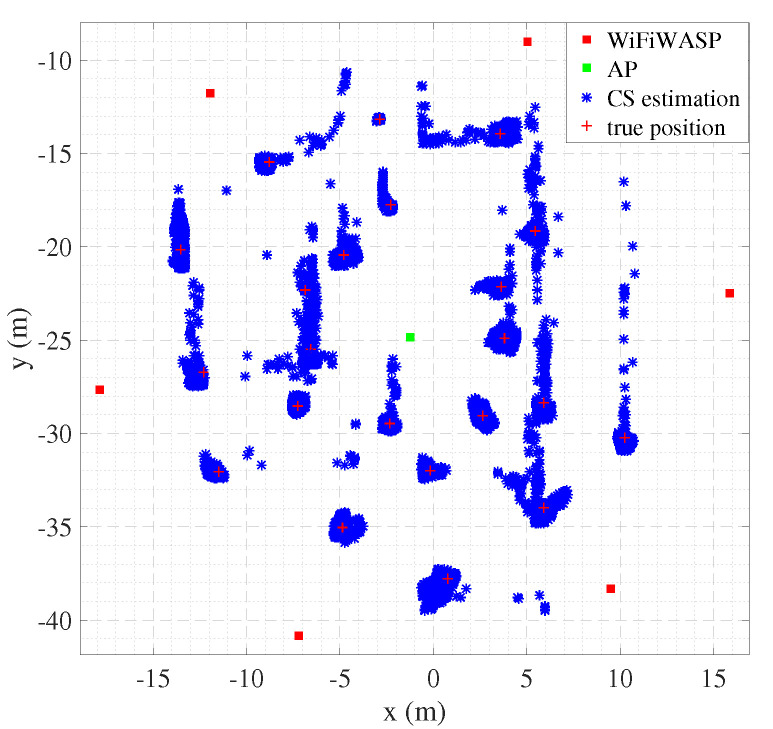
Positioning results in outdoor LOS environments.

**Figure 8 sensors-22-05364-f008:**
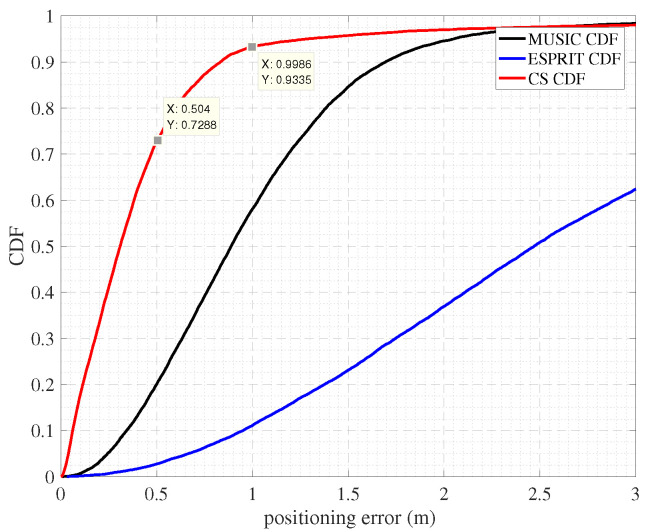
Cumulative distribution function of the positioning errors for the outdoor LOS test.

**Figure 9 sensors-22-05364-f009:**
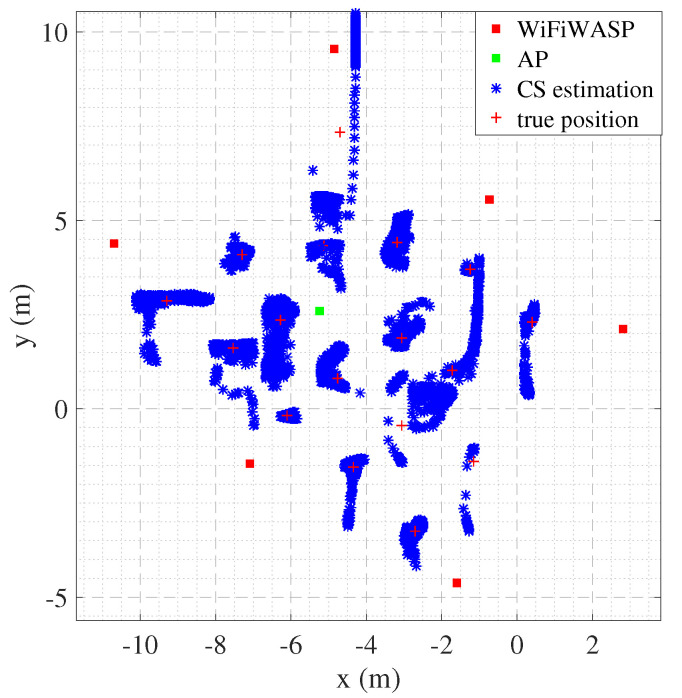
Estimated target locations in the indoor experiment.

**Figure 10 sensors-22-05364-f010:**
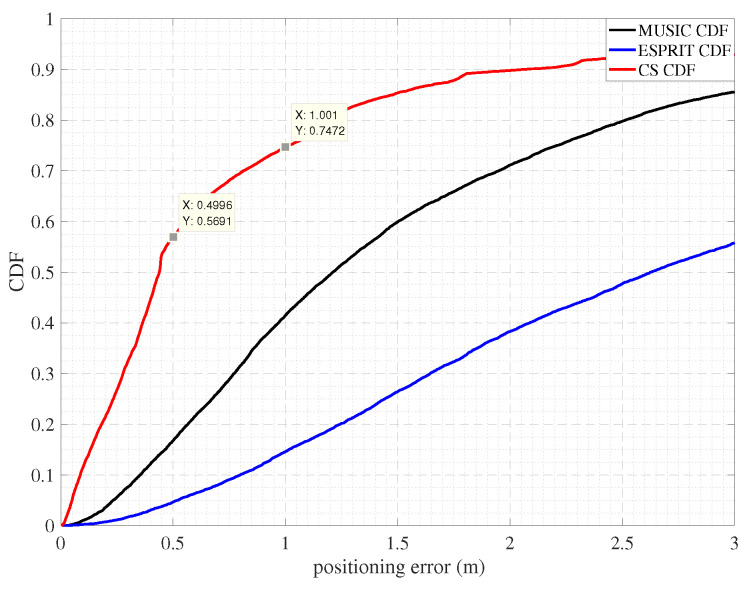
Cumulative distribution function of the positioning errors for the indoor test.

## Data Availability

As the data came from a third party department, CSIRO, which was not authorized to release the data, not applicable.

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
