# Peer review of "The Time-of-Arrival Offset Estimation in Neural Network Atomic Denoising in Wireless Location"

_sensors, 2022, doi:10.3390/s22145364_

Round 1

Reviewer 1 Report

1) In the abstract, the sentence "...for accurate CSI-based positioning accuracy" is a bit contrived. Please rephrase it.

2) The brief review of the existing methods for wireless positioning provided at the beginning of the Introduction misses the important topic of cellular network-based localization. These techniques are experiencing renewed interest thanks to the advent of 5G and 6G technologies. Given their promising features, such techniques will be key enablers for future location-based applications, even in indoor environments. In this respect, I would suggest authors to extend the review by adding some pointers to such a literature. Quite recent and interesting works on the topic that can be added are:

- "Low-Complexity Accurate Mmwave Positioning for Single-Antenna Users Based on Angle-of-Departure and Adaptive Beamforming", ICASSP 2020.

- "Carrier Phase Ranging for Indoor Positioning With 5G NR Signals", IEEE Internet of Things Journal, 2022.

3) Since synchronisation is a very important part of time-based localization systems as the one considered in this contribution, in my opinion authors should extend the part related to "The sniffer measures the time stamp of the communication between the sniffers and the AP to estimate the clock offset and offset of the system clock." by explaining in more details how such a process is actually performed. A more analytical explanation supported by formulas would help to better understand.

4) Page 3, line 104: "Since a 80M bandwidth". I guess authors mean 80 MHz. In any case, the subcarrier spacing should be also specified in order to obtain the total number of subcarriers (256).

5) Before eq. (2), please explain in more details how the CS method operates on (1) to convert CFR into CIR.

6) Some small typos:

- page 3, line 93: "and placed an an access point (AP)";

Author Response

I really appreciate the reviewers’ comments concerning our manuscript. These comments will improve the quality of our paper. we carefully studied reviewer comments and carried on the modification. Therefore, we hope that the new manuscript meets the requirement of the journal. The main corrections in the paper and the responds to the reviewer’s comment are as follows.

Note: All the revisions in the manuscript are in red.

Reviewer 1

  • In the abstract, the sentence "...for accurate CSI-based positioning accuracy" is a bit contrived. Please rephrase it.

Reply: Thanks for your comment. It has been modified in the revised manuscript, please refer to the abstract in the manuscript.

2) The brief review of the existing methods for wireless positioning provided at the beginning of the Introduction misses the important topic of cellular network-based localization. These techniques are experiencing renewed interest thanks to the advent of 5G and 6G technologies. Given their promising features, such techniques will be key enablers for future location-based applications, even in indoor environments. In this respect, I would suggest authors to extend the review by adding some pointers to such a literature. Quite recent and interesting works on the topic that can be added are:

- "Low-Complexity Accurate Mmwave Positioning for Single-Antenna Users Based on Angle-of-Departure and Adaptive Beamforming", ICASSP 2020.

- "Carrier Phase Ranging for Indoor Positioning With 5G NR Signals", IEEE Internet of Things Journal, 2022.

Reply: Thank you for your comments. In the revised manuscript, we have added the reference. Please read introduction in the revised manuscript. Thank you!

3)  Since synchronisation is a very important part of time-based localization systems as the one considered in this contribution, in my opinion authors should extend the part related to "The sniffer measures the time stamp of the communication between the sniffers and the AP to estimate the clock offset and offset of the system clock." by explaining in more details how such a process is actually performed. A more analytical explanation supported by formulas would help to better understand.

Reply: Thank you for your comment. Your comment will greatly benefit the quality of the manuscript. In the revised manuscript, we have carefully modified Section 2.1 in related work. Please read the red section of the related work in the revised manuscript.

4) Page 3, line 104: "Since a 80M bandwidth". I guess authors mean 80 MHz. In any case, the subcarrier spacing should be also specified in order to obtain the total number of subcarriers (256).

Reply: Thank you for your comment. Your comment will greatly benefit the quality of the manuscript. In the revised manuscript, we have carefully modified Section 2.1 in related work. Please read the red section of the related work in the revised manuscript.

Since the system adopts 80MHz bandwidth, there are 256 subcarriers in 121 total, and the subcarrier space is 80MHz / 256 = 312.5khz.

5) Before eq. (2), please explain in more details how the CS method operates on (1) to convert CFR into CIR.

Reply: Thank you for your comment. Your comment will greatly benefit the quality of the manuscript. In the revised manuscript, we have carefully modified Section 2.2 in related work. Please read the red section of the related work in the revised manuscript.

 In the CS method, assuming that the sparse matrix is an identity matrix and there is only rotation factor in the sparse matrix, the multiplication of the sparse matrix by the time domain signal is equivalent to performing the Fourier transform. The physical meaning of sparse matrix is to transform the signal to be sampled into another domain, and in this domain, the signal is sparse.

6) Some small typos:

- page 3, line 93: "and placed an an access point (AP)";

Reply: Thank you for your comment. Your comment will greatly benefit the quality of the manuscript. In the revised manuscript, we have carefully modified Section 2.1 in related work. Please read the red section of the related work in the revised manuscript.

Reviewer 2 Report

This version of the manuscript does not need major corrections before publication. Below, in detail, I described my suggestions that may help to improve this article.

1.        Title

The title should reflect its main idea, e.g., a specific approach, method, scenario, novelty aspect, etc. Generally, the title of the reviewed paper reflects well the paper's contribution.

2.        Abbreviation

Generally, the authors explain most of the used abbreviations, but there are some missing:

·         Page 2; line 46 : abbreviation for CNN is missing

·         Page 2; line 65 : abbreviation for AOA

·         Page 2; line 66 : abbreviation for MUSIC (MUltiple SIgnal Classification)

·         Page 2; line 79 : abbreviation for ESPRIT estimating signal parameters via rotational invariance technique

·         Page 3; line 107 : abbreviation for CFR

·         Page 4; line 112 : abbreviation for ToA – propably TOA – please use the same abbrevaition in the whole paper.

3.        Content

The abstract and introduction provide sufficient background and include relevant references. The research design is appropriate, and the method is adequately described. The conclusions are supported by the results, which are clearly presented.

English language and style are acceptable.

Others:

·         Page 4; line 112 : abbreviation for ToA – propably TOA – please use the same abbrevaition,

·         Page 5: : It woud be nice to have comparison (characteristics) with different kinds of digital filters and then conclusion why the Hamming is choosen,

·         Page 5 – it would be advisable to present the characteristics of the filter used,

·         Page 5; line 134 : Hamming” instead of “Hammin”

4.        References

All references are up to date.

Author Response

Thank you very much for suggestions. These suggestions can help us improve the quality of our manuscript. we have revised the manuscript, according to the comments and suggestions of reviewers and editor, and responded, point by point, the revised contents are listed below.

Reviewer 2

  1. Title

The title should reflect its main idea, e.g., a specific approach, method, scenario, novelty aspect, etc. Generally, the title of the reviewed paper reflects well the paper's contribution.

Reply:  Thank you for your comments.  I revised the title to “ The time-of-arrival Offset Estimation in neural network atomic denoising in wireless location”.

  1. Abbreviation

Generally, the authors explain most of the used abbreviations, but there are some missing:

  • Page 2; line 46 : abbreviation for CNN is missing
  • Page 2; line 65 : abbreviation for AOA
  • Page 2; line 66 : abbreviation for MUSIC (MUltiple SIgnal Classification)
  • Page 2; line 79 : abbreviation for ESPRIT estimating signal parameters via rotational invariance technique
  • Page 3; line 107 : abbreviation for CFR
  • Page 4; line 112 : abbreviation for ToA – propably TOA – please use the same abbrevaition in the whole paper.

Reply:  We are grateful for the problem the reviewer points out. In the revised manuscript, In the revised manuscript, we have defined the abbreviations section 1.

CNN            the Convolutional Neural Network

AOA            angle of arrival  

MUSIC        multiple signal classification

ESPRIT estimating signal parameters via rotational invariance techniques

CFR              the Channel Frequency Response

  As for the abbreviation ToA,  we have revised the wrong abbrevaition in the revised manuscript in Section 2.2. Please read the red section of the related work in the revised manuscript.

  1. Content

The abstract and introduction provide sufficient background and include relevant references. The research design is appropriate, and the method is adequately described. The conclusions are supported by the results, which are clearly presented.

English language and style are acceptable.

Others:

  • Page 4; line 112 : abbreviation for ToA – propably TOA – please use the same abbrevaition,

Reply: We are grateful for the problem the reviewer points out.  we have revised the wrong abbrevaition in the revised manuscript in Section 2.2. Please read the red section of the related work in the revised manuscript.

  • Page 5: : It woud be nice to have comparison (characteristics) with different kinds of digital filters and then conclusion why the Hamming is choosen,

Reply: We are grateful for the problem the reviewer points out. The window function content has been added in Section 3.1. Please read the red section of the related work in the revised manuscript.

  • Page 5 – it would be advisable to present the characteristics of the filter used,

Reply: Thank you so much for your suggestion. In the revised manuscript,  The characteristics of the filters used are described  in Section 3.1.  Please read the red section of the related work in the revised manuscript.

  • Page 5; line 134 : “Hamming” instead of “Hammin”

Reply: Thank you. We have revised the wrong spelling in the revised manuscript in Section 3.1. Please read the red section of the related work in the revised manuscript.

  1. References

 All references are up to date.

Reply: Thank you for your comments. All have been verified and added

Round 2

Reviewer 1 Report

Authors improved the manuscript according to the provided comments.